# Ketogenic Diet and Ketone Bodies against Ischemic Injury: Targets, Mechanisms, and Therapeutic Potential

**DOI:** 10.3390/ijms24032576

**Published:** 2023-01-30

**Authors:** Ciara I. Makievskaya, Vasily A. Popkov, Nadezda V. Andrianova, Xinyu Liao, Dmitry B. Zorov, Egor Y. Plotnikov

**Affiliations:** 1Faculty of Bioengineering and Bioinformatics, Lomonosov Moscow State University, 119992 Moscow, Russia; 2A.N. Belozersky Institute of Physico-Chemical Biology, Lomonosov Moscow State University, 119991 Moscow, Russia; 3V.I. Kulakov National Medical Research Center of Obstetrics, Gynecology and Perinatology, 117997 Moscow, Russia

**Keywords:** ketogenic diet, ketone bodies, ischemia, reperfusion, beta-hydroxybutyrate, acetoacetate

## Abstract

The ketogenic diet (KD) has been used as a treatment for epilepsy since the 1920s, and its role in the prevention of many other diseases is now being considered. In recent years, there has been an intensive investigation on using the KD as a therapeutic approach to treat acute pathologies, including ischemic ones. However, contradictory data are observed for the effects of the KD on various organs after ischemic injury. In this review, we provide the first systematic analysis of studies conducted from 1980 to 2022 investigating the effects and main mechanisms of the KD and its mimetics on ischemia–reperfusion injury of the brain, heart, kidneys, liver, gut, and eyes. Our analysis demonstrated a high diversity of both the composition of the used KD and the protocols for the treatment of animals, which could be the reason for contradictory effects in different studies. It can be concluded that a true KD or its mimetics, such as β-hydroxybutyrate, can be considered as positive exposure, protecting the organ from ischemia and its negative consequences, whereas the shift to a rather similar high-calorie or high-fat diet leads to the opposite effect.

## 1. Introduction

The ketogenic diet (KD) is a high-fat diet with strict limitation of carbohydrate intake. The KD was first established for the treatment of epilepsy [1,2,3,4,5] after Gulep and Marie reported in 1911 the anti-epileptic properties of starvation [6,7], and Wilder in 1921 suggested that the benefits of fasting for seizures might be dependent on ketonemia [8,9,10]. Further studies revealed that the KD may ameliorate some pathological processes, such as inflammation and oxidative stress, affect mitochondria functions, and participate in epigenetic regulation [1]. Based on these findings, nowadays, the KD is used in the treatment of many common pathologies, including cancer [11,12,13,14,15], obesity [16,17,18,19,20,21], neurodegeneration [22,23,24,25,26,27], diabetes [28,29,30], and others.

Recently, the KD was suggested as a therapy for preventing or helping patients recover from ischemic and traumatic injuries [31,32,33,34,35]. Since cardiovascular diseases, including those associated with atherothrombosis and ischemia–reperfusion (IR), are one of the most common “civilization diseases”, investigations of dietary approaches for their correction are of great interest. In 2019, almost 18 million people died from cardiovascular diseases, and 85% of these died due to heart attack or stroke [36], and there is still no effective pharmacological therapy.

Interventions targeting metabolic adaptation can become an important element of supporting therapy, since disturbances in the energy supply of cells during ischemia are a widely recognized element of the pathological cascade [37]. In this regard, active research has been carried out in the last decade on the role of cell bioenergetics modulation in increasing or decreasing the tolerance of organs to ischemic damage. Among such approaches, we can mention a high-fat diet [38], caloric restriction (CR) [39,40], modulation of respiratory activity and mitochondrial function [41,42,43], and some others, including the KD [31,33]. However, so far there has been no systematic review of existing studies on the impact of the KD and its mimetics on IR injury. Thus, in this review we focus exclusively on the mechanisms of the KD and ketone bodies in inducing ischemic damage to various organs.

We used a following search methodology. A basic search was performed on PubMed and Google Scholar. All original research articles from 1980 to 2022 with available abstracts and full texts were included. Reviews, meta-analyses, and case reports were excluded, as well as studies focusing on the effects of interventions that induce endogenous ketosis, such as CR. References in the selected articles were analyzed to identify any other relevant studies that were not included during the previous searches.

## 2. The overview of Ketone Body Metabolism

Before discussing the role of ketone bodies in the pathogenesis of ischemic injury, we provide a brief overview of the metabolism of ketone bodies in the human organism. The extensive topic of ketone body metabolism is beyond the scope of this article and was recently reviewed in the following excellent works [44,45]. In brief, during fasting, exercise, or practicing a KD, fatty acids are relocated from adipocytes to liver cells, and transformed into the Acyl-CoA form with subsequent transport to hepatocyte mitochondria via carnitine palmitoyltransferase 1 (CPT1) to produce ketone bodies, which can be released from mitochondria and then from cells and ultimately into the bloodstream and, instead of glucose, can be oxidized by other tissues to produce ATP (Figure 1).

Ketone bodies include acetoacetate (AcAc), β-hydroxybutyrate (BHB), and acetone. BHB is the most abundant ketone body used as an alternative energy source by extrahepatic tissues in mammals.

In the liver, the enzyme 3-hydroxy-3-methylglutaryl-CoA synthase 2 catalyzes the key reaction for the formation of 3-hydroxy-3-methylglutaryl-CoA (HMG-CoA) from AcAc-CoA, followed by AcAc synthesis. BHB is produced from AcAc in a reaction coupled with the reduction in NAD^+^ to make NADH by the mitochondrial D-β-hydroxybutyrate dehydrogenase (BDH1).

In addition to fatty acids, ketogenic amino acids (such as leucine and lysine) [46] as well as amino acids that are both glucogenic and ketogenic (phenylalanine, isoleucine, threonine, tryptophan, and tyrosine) are involved in ketogenesis [47]. Ketone bodies could also be added exogenously in the form of salts or esters, such as BHB monoester ((R)-3-hydroxybutyl (R)-3-hydroxybutyrate).

Ketolysis is the opposite process to ketogenesis, aimed at the oxidation of ketone bodies in mitochondria. In contrast to ketogenesis, almost all cells (except hepatocytes and some malignantly transformed cells) are capable of ketolysis. The catabolism of ketones proceeds mainly through their conversion into acetyl-CoA, which can either be directed to the tricarboxylic acid (TCA) cycle or used for the synthesis of many substances, including sterols and fats. BHB and AcAc catabolism theoretically yields 22 ATP molecules (5.5 ATP molecules per carbon atom) and 2 GTP molecules (0.5 GTP molecules per carbon atom) per acetyl-CoA molecule during oxidation in mitochondria. Acetone is not converted back to acetyl-CoA and is excreted into urine or via exhalation [48,49,50]. For comparison, for each glucose molecule, 36 ATP molecules can theoretically be produced (6 ATP molecules per 1 carbon atom): 2 ATP due to glycolysis, 2 ATP in substrate phosphorylation in TCA, and 32 ATP due to oxidative phosphorylation (OXPHOS) in TCA reductive equivalents such as flavins and NAD^+^ [51].

According to Mookerjee and colleagues, complete oxidation of glucose yields up to 33.45 ATP molecules per glucose molecule with a maximum P/O ratio of 2.79: 2 ATP molecules per glucose molecule from glycolysis and up to 31.45 ATP molecules per glucose molecule as a result of oxidative reactions during the complete oxidation of glucose to carbon dioxide [52]. Karwi and colleagues noted that the oxidation of each glucose molecule requires 6 O_2_ molecules and forms 31 ATP molecules; the P/O ratio of glucose is 2.58, making glucose the most energy-efficient substrate [53].

The P/O ratio for BHB is 2.50, which makes ketones a less energetically favorable substrate than glucose, but more efficient than fatty acids such as palmitate, whose oxidation requires 23 O_2_ molecules and leads to the formation of 105 ATP molecules with a P/O ratio of 2.33. In addition, the cytosolic processing of the fatty acid into fatty acyl-CoA or triglycerides and the reconversion of triglycerides into fatty acid also consume two inorganic phosphate (Pi) molecules. This further reduces the overall energy gain from fatty acid oxidation.

## 3. The Availability of Ketone Bodies for Extrahepatic Tissues

Ketone bodies are transported into cells of extrahepatic tissues by monocarboxylate transporter (MCT) proteins 1 and 2, also known as solute carrier 16A family members 1 and 7 (SLC16A1; SLC16A7) [54,55]. The mechanism by which BHB enters mitochondria is still unknown; however, the mitochondrial pyruvate carrier (MPC) is suggested to be one of the most probable ways [56] (Figure 1). In mitochondria, BHB is converted to AcAc by BDH1 and then succinyl-CoA:3-ketoacid CoA transferase (SCOT) catalyzes a reversible and near-equilibrium conversion of AcAc to AcAc-CoA (Figure 1).

Ketone bodies are able to cross the blood–brain barrier (BBB) through MCTs in endothelial cells and astroglia [57]. With increased generation of ketone bodies, a linear relationship between the concentration of BHB in plasma and its transport through the BBB exists, according to data obtained in fasted rats [58]. Prolonged fasting appears to increase the transport of ketones through the BBB, with an 8-fold increase in the expression of MCT1 [59]. Similarly, a 13-fold increase in cerebral uptake of BHB in humans following short-term starvation was shown [60]. In contrast, cerebral ketone concentrations following a rapid intravenous infusion of BHB increase only slightly compared with in long fasting. This possibly implies that MCT upregulation is dependent on the duration of exposure to increased plasma ketone levels [61,62].

Physiological levels of ketone bodies and their transporters vary in different periods of development [63]. In particular, the level of ketone bodies is highest during brain development and declines in adults, with a peak at 48 h after birth [64]. The adult human brain consumes 20–23% of the body’s daily calorie intake, while it accounts only for 2% of body weight. In newborn infants, the brain represents about 11% of body weight but consumes at least 60–70% of the infant’s calorie intake, so additional sources of energy are especially required in the early stages of development [65,66].

It is also important to note that tissues differ in their ability to consume and produce ketone bodies. Thus, the ketolytic capacity of tissues is proportional to their level of SCOT activity. For example, the human liver expression of the SCOT gene is almost completely suppressed [67,68], making the liver only capable of producing ketone bodies, which is probably a mechanism to avoid futile cycling. The heart and kidney are known to have high SCOT activity, thus having the greatest capacity for ketone body utilization [69]. Notably, different rat hepatoma cell lines sometimes show SCOT expression, with SCOT being detected at low levels in normal rat hepatocytes [70]. Extrahepatic tissues have been shown to primarily consume ketone bodies rather than produce them.

However, there is some evidence that several cell types besides hepatocytes might be able to produce ketone bodies; for example, BHB is also synthesized in astrocytes, which are the only type of brain cell capable of oxidizing fatty acids [71,72,73], and can supply other types of brain cells using BHB as fuel. This echoes the principle of the lactate astrocyte–neuron shuttle, which in brief proposes that astrocytes can produce lactate, which can be taken by neurons and used as a fuel in some conditions [74]. Perhaps the same is true for ketone bodies as well [75].

Moreover, in mammals, only hepatocytes and colon epithelial cells abundantly express the mitochondrial isoform of 3-hydroxy-3-methylglutaryl-CoA synthase-2 (HMGCS2) [76], indicating that the colon may also be a ketogenic organ. In addition, overexpression of the *Hmgcs2* and an increase in HMGCS2 levels in the kidney under some conditions, such as starvation or diabetes type 2, indicate the potential for ketogenesis to occur in the kidney [77,78].

## 4. Therapeutic Ketogenic Interventions

### 4.1. Types of KD

There are several types of KD, and all of them are characterized by a very low carbohydrate content, with lipids and protein being the main source of energy. The most common KDs are the classic KD, the medium-chain triglyceride diet (MCTD), the modified Atkins diet (MAD), low glycemic index treatment (LGIT), and the very low calorie KD (VLCKD) [79,80].

The classic KD was first described in 1921 [8] and assumes a 4:1 ratio of fat to carbohydrates plus protein, and 90% of calorie intake from fat. The KD was traditionally started in the hospital after a 48 h fast, followed by a gradual introduction of the KD over 3 days [81,82]. This is a strict diet that requires medical supervision, because it has a number of absolute and relative contraindications, as summarized earlier [1,83]. The classic KD, lasting from 1 month to 5 years, can also lead to side effects such as dehydration, hypoglycemia, metabolic acidosis, weight loss, high levels of low-density lipoprotein, elevated total cholesterol, constipation, diarrhea, vomiting, abdominal pain, and nephrolithiasis, but potential side effects can be avoided with care during the diet’s initiation and maintenance phases [1,84].

The MCTD is another type of KD, mainly consisting of octanoic (C8) and decanoic (C10) fatty acids, which yield more ketones per kilocalorie of energy compared to the classic KD. The high ketogenic potential of the MCTD allows less total fat intake (30–60%) and inclusion of more carbohydrates (15–19%) and protein (10%), which gives a 1:1 or 2:1 ratio of fat to carbohydrates and protein. This makes the MCTD more favorable and palatable [85,86,87,88].

The MAD was first introduced in 2003 [89] and is similar in fat composition to a 0.9:1 ratio in grams with approximately 65% of the calories ingested coming from fat, 25% from protein, and 10% from carbohydrates [90]. The MAD is considered to be more flexible and palatable than the classic KD.

The LGIT, introduced in 2005, requires a less stringent carbohydrate restriction than earlier KDs [91]. This alternative diet is based on a 0.6:1 ratio of fat to carbohydrates and protein, containing 60% fat, 30% protein, and 10% carbohydrates with a glycemic index lower than 50.

The VLCKD mimics fasting through a restriction of calorie intake. The VLCKD includes different stages, starting with significant caloric and carbohydrate restriction for approximately 12 weeks (<700–800 kcal/day). Thus, the first stages of the VLCKD are characterized by a low carbohydrate content (~13% of total energy intake), and relatively high proportions of fat (~44%) and protein (~43%) [80,92,93,94].

In addition, there is another classification of KDs that is more popular among athletes. This classification includes the following KDs: the standard KD, cyclical KD, targeted KD, and high-protein KD (with approximately 35% of ingested calories coming from protein) [95]. The cyclical KD alternates periods of KD adherence with periods of high-carbohydrate consumption. The targeted KD is pretty similar; it allows more carbohydrates after an intensive physical workout.

Although any KD implies an increased intake of fat, it is important not to confuse it with the Western diet (WD), implying its origin from developed Western countries. This type of diet includes easy meals containing large amounts of processed meats, saturated and/or trans fats, sodium, and sugars. Such meals have a high caloric value within a small portion, but contain small amounts of vegetables, vitamins, and unsaturated fats. The typical WD is excessively high in fat and sugar [96]. A high-calorie WD combined with chronic overeating is closely related to the Western lifestyle, including low levels of physical activity, insufficient sleep, increased psychological stress, smoking, etc. These mismatches between humans’ normal physiological needs and the WD and lifestyle, combined with a prolonged lifespan, have contributed to the development of so-called «civilization diseases» such as obesity, diabetes, cardiovascular diseases, and cancer [97,98].

### 4.2. Association of KD and CR

There is an association between the effects of the KD and CR. CR is another dietetic treatment that has demonstrated beneficial effects in therapy of various pathologies, including type 2 diabetes, cardiovascular disease, and cancer, and ultimately as an approach for increasing longevity [99,100,101]. The protective effect of CR on different models of kidney injury in rodent studies [102], including IR [103,104], was revealed. CR has been proven to be an effective therapeutic approach in rodent models of ischemic injury of the brain, liver, gut, and heart [105,106,107,108,109].

The positive effects of CR could be due to the increase in circulating concentrations of ketone bodies, which indeed occurs during any fasting state [110,111]. What is discussed less is the fact that the KD can be accompanied by CR itself. The KD is proposed as an approach for obesity treatment because it effectively reduces appetite [112,113,114]. The exact molecular mechanisms by which KDs suppress appetite remain to be established. It is hard to determine exactly whether the appetite suppression seen with ketogenic diets is indeed due to ketosis, or due to other factors such as an increased or decreased content of different nutrients. However, the KD and the administration of exogenous ketones appear to decrease ghrelin secretion and reduce hunger [115,116]. Thus, animals and people on a KD might have their calorie intake lowered by loss of appetite, so they are both in KD and CR states. This poses quite a challenge from a methodological perspective; *ad libitum*-fed animals on a KD might be in a CR state due to their own appetite, while *ad libitum*-fed animals on a standard diet will not be. This methodological problem was also discussed in [117], and the authors even attributed the KD to low-calorie dietary interventions. Sometimes, calorie intake in the control group was compared with that of the KD group. These cases should be carefully evaluated to allow such matching.

On the other hand, KD food is highly caloric (standard laboratory food has 300 kcal/100 g, while ketogenic food has 600), and if for any reason an animal’s appetite is not suppressed, they might consume more calories and gain weight [118,119]. Our analysis has shown that only in 7 of 22 KD (Table 1) and 1 of 21 KD mimetics (Table 2) studies was such control of calorie intake ensured [31,32,33,120,121,122], and only 11 of 22 monitored weight [31,32,33,34,118,119,123,124,125,126,127]. Thus, in some studies, KD animals could be considered as KD-CR, and in others, KD-high calorie, and it is impossible to verify this without direct calorie intake measurements. This should be kept in mind while interpreting results.

### 4.3. KD Mimetics

KDs and derivative diets have limitations in their use. The MAD and LGIT diets or avoiding the fasting step could prevent KDs’ side effects [152] but evoke an adverse lipid profile [153], severe and repeated vomiting, and various gastrointestinal problems [154,155]. In addition, for many patients, it is hard to maintain a specific diet consistently. Thus, searching for potential pharmacological mimetics is very important for clinical practice.

The most obvious choice of KD imitation is the use of ketone bodies themselves. BHB is conventionally used both in animal studies and in vitro experiments with cell cultures (Table 2). Among 21 articles describing the use of KD mimetics to reduce IR injury, 18 studies considered BHB (usually D-BHB sodium salt is used, although in three studies DL-BHB isoform was administrated [138,141,146]). Only in two studies was AcAc used—as part of a mixture with BHB [144] or alone in the form of lithium salt [145].

Among other mimetics are ketone body precursors. These include 1,3-butanediol metabolizing in the liver to ketone bodies [149,156] and esters such as (R)-3-hydroxybutyl-(R)-3-hydroxybutyrate and R,S-1,3-butanediol AcAc diester [157,158,159,160]. Acute ingestion of either ketone ester leads to a short-term (from 0.5 to 6 h) nutritional ketosis, indicated by a serum BHB concentration increase over 1 mM [161,162]. Nutritional ketosis in this case is achieved without fasting or a KD and prevents problems with salt or the non-bioactive form of BHB overloading; those will be discussed below in the “Limitations of KD Mimetics Administration” section.

Medium-chain triglycerides (MCTs) could be used as KD mimetics as well. Mentioned earlier in the context of the MCTD, MCTs are rapidly taken up by the liver and converted into ketone bodies. Ketone esters and MCTs have not been used in KD mimetics studies associated with IR injury yet, but in one study, 60% of the carbohydrate calories were replaced with one or more of the following substrates: 1,3-butanediol, triacetin, tributyrin, long-chain triglycerides (LCTs), and MCTs [121].

### 4.4. Limitations of KD Mimetics Administration

The usage of ketone bodies in animal studies has some limitations, which must be considered.

#### 4.4.1. Alterations in Macronutrients Intake

The most important thing to consider is the carbohydrate proportion. In fact, the administration of ketone bodies usually occurs in a normal carbohydrate diet. However, it has been quite widely proposed that the main effects of a KD come from carbohydrate restriction rather than from ketone bodies; thus, this approach misses one of the most crucial features of a KD. For example, significant infarct volume reductions were observed in rats fed nasogastric diets with 1,3-butanediol and triacetin–tributyrin instead of carbohydrates, wherein the volume of the infarct was directly related to the plasma glucose concentration before ischemia, but not to plasma ketone body levels [121]. Low-glucose conditions stimulating a KD caused ATP efflux from pyramidal neurons in the CA3 hippocampus that induced conversion of ATP to adenosine with subsequent activation of inhibitory adenosine receptors, coupled with K_ATP_ channel activation, providing an anticonvulsant effect of the KD [163].

Secondly, the effects of a KD may be due to changes in the ratio of certain nutrients, which is not taken into account when ketone bodies or their precursors are used. For example, some effects of the CR diet may be due to the restriction of essential amino acids such as tryptophan and methionine [164]. The KD seems to play an important role in amino acid metabolism [165,166] and results in decreased aspartate production and enhanced glutamine synthesis [165].

The third point is the adverse effects of using salts or acids in quantities necessary for stimulation of a KD. Achieving a fasting or KD-like concentration of blood BHB salts would require the consumption of high levels of sodium. In theory, a balanced mineral combination of ketone salts containing multiple electrolytes may be useful in attenuating symptoms of mineral depletion that occur early in response to a KD. However, excessive mineral intake from ketone salt supplements can lead to gastric hyperosmolarity along with potential gastrointestinal distress [167,168]. Moreover, most ketone salts are racemic, with 50% including the bioactive D-isoform of BHB and 50% the L-isoform of BHB [158], which is not normally the result of ketogenesis. Among 18 studies investigating the effects of BHB on IR injury, only 8 used D-BHB (Table 2) [120,135,140,142,143,146,147,148].

#### 4.4.2. The Influence on Microbiota

Another important thing that is often overlooked in the use of KD mimetics is the alteration of the microbiota. Changes in the gut microbiome are observed in children with epilepsy after 1 week on the KD, although its association with improved epilepsy symptoms is not clear [169]. Later, the anticonvulsant properties of the KD in mice mediated by changes in specific bacterial species in the gut were demonstrated [170,171]. However, it is not completely clear whether the KD mimetics have the same effect on the microbiota.

#### 4.4.3. The Difference in KD and Mimetic Treatment Duration

It must be kept in mind that ketosis caused by the KD is usually chronic; humans with epilepsy might stick to it for ages, while laboratory animals are maintained on it for weeks and rarely months. However, this allows us to consider all possible mechanisms, including metabolic shifts, changes in gene expression, epigenetic regulations, and so on. However, in some studies, mimetics are used in low doses or for short periods: one bolus shortly prior to or shortly after injury [137,138,148]. Of course, this approach is perfectly legitimate for scientific use, but it obviously may miss some complex effects of the KD, which can be both positive and, more important, negative in the case of ischemic injury and other pathologies. Maximum care should be taken in interpreting results and making medical recommendation for KD use based on the results of KD mimetics studies.

#### 4.4.4. Cell Model Limitation

It is necessary to pay attention to some features of the metabolism of ketone bodies, which are important for in vitro experiments. Cell cultures can be considered to belong to four different types: (1) originating from various cancers; (2) immortalized cell lines; (3) primary cell lines from embryos or newborns; (4) primary cell lines from adult organisms. Different cells may have different nuances of ketone metabolism. Thus, embryonic cells might have increased ketone metabolism, since embryonic tissues rely on ketone metabolism more than adult organisms [63], while in cancer cells, ketone metabolism can be impaired [172]. These effects are related to the Warburg effect or aerobic glycolysis [173]. Cancer cells preferentially use glycolysis for energy, regardless of the availability of oxygen. This leads to the production of fewer ATP molecules per glucose molecule, so tumor cells require a large amount of glucose. This transition from oxidative phosphorylation to glycolysis and a decrease in TCA cycle activity appear very early on in oncogenesis and are one of the hallmarks of cancer [174]. It is believed that the metabolic changes found in cancer cells reduce their ability to shift the primary energy source [175,176,177,178]. By reducing the availability of glucose for cancer cells and providing ketones as an alternative energy source for normal cells, the KD may limit the growth of highly glycolytic tumors such as malignant gliomas.

## 5. Effects of KD and Ketone Bodies on Ischemic Damage

We listed available studies exploring the effects of either KDs and high-fat diets (Table 1) or KD mimetics (Table 2) on ischemic injury. We paid attention to some important nuances of these works, such as the duration of administration of ketone bodies or calorie intake normalization during a KD. Approximately half of the studies are related to brain ischemia, and the rest reveal the effects of ischemia on other organs (heart, gut, kidney, liver, eyes) or in vitro models of IR injury. This section provides a brief overview of existing experimental models used to study the effects of the KD and its mimetics on IR damage to the brain, kidney, and heart.

### 5.1. Cerebral IR Injury

The KD, 3–4 weeks prior to brain ischemia, demonstrated a protective effect on the middle cerebral artery occlusion (MCAO) model, reducing the infarct volume size [33,34,35,121,122,129,135]. The KD also ameliorated ischemic stroke induced by endothelin-1 injection [130] and neurodegeneration caused by cardiac arrest global ischemia [134]. Non-standard KDs, e.g., the “new-KD”, the soybean oil diet, and the triheptanoin-rich diet, protected the brain from traumatic injury and MCAO, facilitating systemic ketosis [32,122]. Experimental diets based on 1,3-butanediol, triacetin, and tributyrin reduced cerebral infarct, but mostly by lowering the plasma glucose concentration before MCAO [121].

Similar protective effects have been shown for KD mimetics, mostly after their short-term administration. BHB administration before or after injury ameliorated both permanent and transient MCAO, endothelin-induced ischemia, N2-induced hypoxia, and KCN-induced anoxia [136,137,141,144,147,148,151]. Relatively long-term administration of BHB was also protective against MCAO [135] and promoted recovery from spinal cord injury (SCI) [143]. Other KD mimetics, such as 1,3-butanediol and AcAc, had a beneficial impact during microsphere-induced ischemia [149] and were excitotoxicity induced by iodoacetate and L-trans-pyrrolydine-2,4-dicarboxylate [145].

In rat cortical neuronal cells, in vitro treatment with 5–20 mM BHB alone did not affect cell viability, but protected against neuronal cell damage after oxygen glucose deprivation (OGD) [137]. Pretreatment of neuroblastoma cells with 10 mM BHB reduced OGD-induced cell death [33,35]. Transfection with small interfering RNA (siRNA) targeting the A1 adenosine receptor before BHB pretreatment reversed the protective effect of BHB, indicating the important role of adenosine signaling pathways in brain ischemic tolerance [35]. OGD and reperfusion caused significant mitochondrial fragmentation, whereas 10 mM BHB prevented the mitochondrial fission [33].

### 5.2. Renal IR Injury

In the case of renal IR, only short-term ketogenic interventions were studied. The administration of a KD for 3 days had a nephroprotective effect on the model of ischemic injury of the kidney, reducing the number of injured tubules and preserving kidney function [31]. Treatment with BHB via an osmotic pump (24 h before injury and 24 after) had protective effects on mice, preserving kidney function and reducing the number of apoptotic cells in the tissue [139]. In vitro, it was shown that 1 to 10 mM BHB had little effect on the cell viability of HK-2, proximal tubular cells from normal kidneys, although cell viability significantly decreased in normoxic HK-2 cells after treatment with 20 and 40 mM BHB. However, the treatment with 10 mM BHB significantly improved the cell viability of hypoxic HK-2 cells [139].

### 5.3. Myocardial IR Injury

For the heart, a long-term low-carbohydrate KD significantly improved recovery after ischemia [124], as well as BHB administration a few hours before or after an ischemic insult reduced infarct size and apoptosis [142,146,150]. However, in a few articles, negative effects of high-fat diets or its mimetics on IR injury were demonstrated. For instance, a 4-month KD decreased mitochondrial biogenesis, reduced cell respiration, and increased cardiomyocyte apoptosis and cardiac fibrosis in rats [179]. Exogenous BHB administration mimicked these effects and affected the sirtuin 7 promotor.

Interestingly, most of the reported adverse effects of the KD on in vivo models were associated with heart damage. To understand this inconsistency, a series of studies by Jian Liu were performed focusing on the effects of a high-fat and low-carbohydrate (HFLCD; 60%—fats, 10%—carbohydrates) dietary pretreatment before ischemic heart injury [118,119,132]. It was hypothesized that the adverse effects of the HFLCD on myocardial IR injury may be mediated through enhancing oxidative stress and mitochondrial damage [119]. The HFLCD inhibits mitochondrial fusion and increases fission following IR [118]. Of note, the time at which the diet is given may be responsible for the observed effects. Introducing a high-fat diet before myocardial infarction may be harmful, while the same diet consumed after myocardial IR may cause beneficial effects [125,126].

## 6. Possible Mechanisms of the Beneficial Effects of KD on IR injury

This section summarizes the possible mechanisms of the beneficial effects of the KD and ketone bodies on the treatment of IR injury (Figure 2).

### 6.1. Induction of Anti-Inflammatory Response

Acute ischemic injury initializes a pro-inflammatory response to the presence of dead cell debris in the injured zone. Reperfusion exacerbates this pro-inflammatory response, for example, it is well-known that myocardial IR injury manifests between 6 and 24 h post-reperfusion [180,181,182]. Induction of inflammation due to IR injury was also shown for the brain and kidneys [183,184,185]. A pro-inflammatory response is followed by an anti-inflammatory reparative phase. Perturbations in both the balance and transition between the pro-inflammatory and the anti-inflammatory phases can enhance acute IR injury and its negative consequences.

Cell death causes the release of nucleic acid fragments, which can be recognized as damage-associated molecular pattern molecules (DAMPs). A wide array of cellular stressors, including IR injury, can lead to the DAMPs release [186]. DAMPs bind to pattern recognition receptors (PRRs), which trigger a cascade of inflammatory mediators. Downstream signaling of the PRRs leads to activation of the mitogen-activated protein kinases (MAPKs) and nuclear factor-κB (NF-κB) pathways, as well as the NOD-, LRR-, and pyrin domain-containing protein 3 (NLRP3) inflammasome. These pathways regulate the expression of a range of genes including pro-inflammatory cytokines (tumor necrosis factor-α (TNF-α), IL-1β, IL-6, and IL-18) and chemokines. While leukocytes amplify the inflammatory response, they also promote efferocytosis of dying cells and tissue, participating in the transition to inflammation resolution [187,188,189,190].

The protective effects of the KD and its mimetics on animal models often correlate with reduced pro-inflammatory signaling and are characterized by decreased expression of the NF-κB and reduced levels of pro-inflammatory molecules such as IL-1β, IL-6, IL-18, TNFα, IFNγ, and MCP-1 [31,33,138,139,143].

Additionally, ischemia is known to cause endoplasmic reticulum (ER) stress caused by the reduced protein folding capacity of the ER, thus leading to the accumulation of misfolded proteins [191,192,193]. ER stress can activate the NLRP3 inflammasome [194], which leads to an inflammatory response. It has been demonstrated that pretreatment of SH-SY-5Y cells with 10 mM BHB significantly inhibits NLRP3 expression, IL-1β release, and caspase-1 activity caused by ER stress after OGD. In mice exposed to MCAO, a 3-week KD attenuated ER stress by PERK-eIF2-ATF4 pathway inhibition, which enhanced the expression of important chaperone proteins involved in ER stress, such as GRP78 and HSP70. The expression of ER-specific apoptotic components C/EBP homologous protein (CHOP) and cleaved caspase-12 was reduced, while pro-caspase-12 expression increased in KD-fed mice with MCAO [33]. One-day subcutaneous administration of BHB after left anterior descending coronary artery occlusion in mice reduced levels of GRP78, CHOP, and X-box-binding protein 1 (XBP-1), indicating attenuation of ER stress [142].

The inflammatory response following activation of NLRP3 inflammasome plays a significant role in cell death during ischemic stroke and in some pathological conditions, such as SCI [195,196,197], and BHB inhibited NLRP3-associated inflammation [143,198,199]. Activation of the TXNIP-NLRP3 pathway in the brain of MCAO mice was accompanied by raised caspase-1 activity, while IL-1β levels were decreased in mice kept on a KD and after supplementation with BHB in vitro [33]. Ischemia-mediated NLRP3 inflammasome activation was attenuated by BHB administration in warm hepatic IR injury, resulting in decreased serum alanine aminotransferase (ALT) and IL-1β levels [138].

### 6.2. Attenuation of Oxidative Stress

Oxidative stress in IR-injured tissues is caused by mitochondrial dysfunction and subsequent production of reactive oxygen species (ROS), reactive nitrogen species (RNS), and reactive electrophile species (RES), associated with neuronal, myocardial, hepatic, and renal cell death [200,201,202,203]. Treatment with a KD or its mimetics is associated with decreased markers of oxidative stress in animal models. Particularly, a 3-week HFLCD resulted in lower levels of inducible nitric oxide synthase (iNOS) and NO production in murine gut 1 h after ischemia and 3 or 6 h after reperfusion [131]. In vitro, pretreatment of SH-SY-5Y cells with 10 mM BHB decreased ROS production after OGD [33], and intraperitoneal injection of 500 mg/kg BHB 1 h after a stroke reduced ROS production in rats’ neocortex [141]. The KD diminished DNA/RNA oxidative damage indicated by 8-hydroxy-2′-deoxyguanosine (8-OHdG) or 8-hydroxyguanosine (8-OHG) immunostaining in several IR models [31,32].

The effects of the KD on lipid peroxidation are controversial: the KD prevented an increase in RNS and electrophilic stress markers such as 3-nitrotyrosine (3-NT) and 4-hydroxynonenal (4-HNE, lipid peroxidation end-product) after IR [31], but induced transient increases in H_2_O_2_ and 4-HNE in rat brains [204]. However, the increase in H_2_O_2_ seemed to stimulate the nuclear respiratory factor 2 (Nrf2) antioxidant pathway through Nrf2-Keap1 redox sensing, since the growth in Nrf2 levels was detected in the brain and liver of rats kept for 1 week on a 78% fat-containing KD [204,205]. Nrf2 promotes the expression of antioxidative genes, including heme oxygenase-1 (HO-1), SODs, glutathione peroxidase (GPX), CATs, and NAD(P)H:quinone oxidoreductase-1 (NQO1), providing a protection of cells against cerebral, liver, myocardial, and renal ischemic injury [206,207,208,209]. Fasting or exogenous injection of BHB was accompanied by nuclear translocation of Nrf2 in the retina, which led to the protective effect on retinal degeneration after ischemia induced by the optic nerve and central retinal blood vessel transection [120]. This effect of BHB was explained by the elevation in retinal fumarate, an intermediate of the TCA cycle, through the enhancement of ketolysis. However, in IR-induced kidney injury in rats, the KD increased only the activity of the antioxidative enzymes GPX and SOD, but not CAT [31], while subcutaneous administration of BHB reversed the reduction in CAT mRNA expression in an SCI model [143]. By contrast, after IR of isolated hearts, antioxidative SOD2, CAT, and GPX1 levels were reduced in an HFLCD-fed group, along with increased levels of pro-oxidative NADPH oxidase-4 and xanthine oxidase [119].

### 6.3. Epigenetic Regulation

Epigenetic modifications of DNA and histones can serve as sensitive indicators of transcriptional changes. Recently, an increasing number of studies have pointed to the important role of epigenetic modifications in response to IR injury [210,211,212,213]. Histone acetylation and histone deacetylase (HDAC) activity is one of the most extensively studied epigenetic marks [214], and HDAC inhibitors are considered potential therapeutic agents for neurodegeneration associated with ischemic stroke [215,216,217], renal, and cardiac IR injuries [218,219,220,221,222]. However, despite their renal, neuro-, and cardioprotective actions, HDAC inhibitors are known to cause a wide range of adverse effects [223,224]. For this reason, there is growing interest in BHB as a potential inhibitor of HDAC. Indeed, BHB has HDAC inhibitory activity even at a low concentration level (1–2 mM) [225]. In turn, histone acetylation upregulates many genes associated with antioxidative and anti-inflammatory pathways. BHB induces histone acetylation at the promoter of transcription factor forkhead box protein O (FoxO) [226] by inhibiting HDAC classes I and II [225,227]. FoxO transcription factors play crucial roles in ROS suppression, induction of apoptosis, engagement of autophagy and regulation of cell metabolism [228,229,230]. FoxO1 is the most widely studied FoxO transcription factor, playing a neuroprotective role in ischemic stroke [231].

One of the FoxO1-regulated genes is HO-1, an inducible enzyme responsible for the breakdown of heme that has antioxidative and anti-inflammatory properties [232]. Both an injection of 10 mmoL/kg of exogenous BHB and a rise of serum BHB caused by 12 h of fasting induced histone-3 acetylation, which resulted in higher expressions of FoxO1, followed by HO-1 upregulation in mice [138]. Remarkably, the addition of glucose to drinking water reversed the BHB-induced increase in HO-1 and FoxO1 protein levels after liver ischemia [138].

Similarly, FoxO3 plays an important role in apoptosis and autophagy [233], regulates adaptive responses that counteract the adverse effects of chronic hypoxia on the kidney [234,235,236], and is associated with NLRP3 inflammasome-mediated endothelial cell pyroptosis [237]. It has been shown that levels of FoxO3 were significantly lowered in IR-injured kidneys in mice, while treatment with BHB reversed this effect [139]. Additionally, 10 mM BHB ameliorated FoxO3 decreases in hypoxic and H_2_O_2_-treated HK-2 cells [139], suggesting renoprotective effects of BHB that are mediated by FoxO3 upregulation and increased H3K9 acetylation. A decrease in apoptosis repressor with caspase recruitment domain (ARC) and an increase in caspase-1, IL-1β, and IL-18 in IR-injured kidneys and in hypoxic or H_2_O_2_-treated HK-2 cells was reversed by BHB. Moreover, treatment with BHB suppressed SCI-induced NLRP3 inflammasome activation and reduced protein levels of caspase-1, IL-1β, and IL-18 and mRNA expression of FOXO3a, indicating a possible anti-pyroptotic effect of BHB [143].

Besides the classes I or II of deacetylases, in mammals, seven members of the sirtuin protein family, known as class III histone deacetylase, have been identified. Among the sirtuin members, the NAD^+^-dependent mitochondrial deacetylase sirtuin-3 (SIRT3) has received much attention for its role in the regulation of cellular metabolism, including the adaptation to stresses such as fasting, CR, or exercise. SIRT3 protein levels are upregulated by CR in the liver, where it stimulates fatty acid oxidation and ketogenesis. It is well known that SIRT3 mediates ROS elimination and apoptosis prevention [238,239,240] by activating FoxO3a followed by induction of SOD2 [241] and CAT to reduce ROS [242,243]. It was hypothesized that SIRT3 could mediate the neuroprotective effects of ketone bodies after ischemic stroke [144] through a BHB-mediated increase in NAD^+^ levels and subsequent NAD^+^-dependent sirtuin activation [244]. BHB and AcAc injection 30 min after MCAO led to enhanced mitochondria function, reduced oxidative stress, reduced infarct volume, and improved neurologic function in mice, while SIRT3 knockdown in vitro diminished ketones’ beneficial effects [144]. Note, the SIRT-FoxO axis is also activated in response to moderate CR [228,245], suggesting the effects of the KD on FoxO-dependent regulation could be mediated not only by ketone bodies.

### 6.4. Energy Supply Restoration and Metabolic Adaptations

The preferred source of brain energy is glucose. However, even under normal conditions, the brain also metabolizes alternative substrates, such as pyruvate [246], lactate [247], acetate [248], and BHB [249]. The addition of substrates (BHB, AcAc, pyruvate) to cells did not increase the basal oxygen consumption rate (OCR) in neurons or mixed glia. However, simulating the conditions of maximum energy consumption with the mitochondrial uncoupler FCCP induced an increased OCR, where pyruvate gave the highest OCR value, and AcAc and BHB caused a modest increase in the OCR. Thus, both neurons and glia are able to use ketone bodies as alternative energy sources [250]. Most of the BHB oxidized in the brain comes from the liver via the bloodstream. BHB is also synthesized endogenously in astrocytes, which are the only type of brain cell capable of oxidizing fatty acids [71] and supply other brain cell types with BHB as fuel.

If under normal conditions ketone bodies are an additional source of energy, then under certain pathological conditions they may provide the only chance for cells to survive. The simplest possible mechanism for the neuroprotective effects of the KD is an energetic role of ketone bodies maintaining mitochondrial function, ATP production, and neuronal survival. It is worth noting that the pathogenesis of ischemic stroke includes insulin resistance [251], leading to inhibition of pyruvate dehydrogenase activity, a key step in the conversion of pyruvate to acetyl-CoA and functioning of the TCA cycle. The metabolism of ketone bodies provides a readily available source of acetyl-CoA, bypassing the blockade of the pyruvate dehydrogenase complex.

During low-energy conditions, such as ischemia, the AMPK pathway is activated. This is a signal to reduce ATP consumption and enhance its production. Activation of AMPK is known to be protective against IR injury of the heart, brain, kidneys, or guts through decreasing the cellular ATP consumption and the need for oxygen, resulting in enhanced cell survival [252,253,254,255,256]. Indeed, adherence to a KD for 4 weeks before MCAO induced a significant increase in the protein levels of AMPK compared to a group kept on a normal diet, as well as an increase in hypoxia-inducible factor-1α (HIF-1α) levels [34], regulating cellular metabolism adaptation to hypoxia [257,258,259]. In this way, a 3-week KD increased expression of HIF-regulated genes such as EPO, VEGF, *GLUT-1*, and *MCT-4* after brain ischemia in mice [35], promoting angiogenesis, cell proliferation and migration, increased glucose metabolism, and cell survival. An increase in HIF-1α protein in the rats fed the KD or treated with BHB was demonstrated, and such an increase was also observed with the high-carbohydrate-diet group as well as the 3-NPA or propionate instead of BHB infusion [135]. Additionally, AMPK and HIF-1 regulate the induction of autophagy [260,261,262,263,264], and treatment with BHB reduced the ratio of two forms of microtubule-associated protein 1A/1B-light chain 3 (LC-3)—LC3-II/LC3-I, and protein levels of p62, elevated in the myocardium of IR-injured mice. Moreover, BHB administration upregulated protein levels of Lamp2, a marker of lysosome function, in the myocardium of IR mice [142].

The utilization of ketones may result in an increase in intracellular succinate, a known inhibitor of prolyl-hydroxylase (PHD), the enzyme responsible for the degradation of HIF-1α. Indeed, succinate elevation after both a KD and BHB infusion was shown along with a decrease in PHD mRNA expression [135]. Contrariwise, there were no significant changes in succinate levels in the KD group reported by Xu and colleagues. [34]. The effects of the KD on HIF-1α may also mediate the effect of the diet on fibrosis, since HIF is well known as a modulator of fibrosis [265,266,267]. The development of fibrosis caused by 30 min of left kidney ischemia in rats was reduced by 3 days of KD feeding before surgery [31].

### 6.5. Effects on Mitochondria

In some cases, the triggering of apoptotic or necrotic cell death is preceded by the opening of the mitochondrial permeability transition pore (mPTP) [268,269,270,271]. Activation of the mPTP leads to mitochondrial dysfunction [272,273,274,275], swelling of mitochondria [276], mitochondrial fragmentation [277], and rupture of the outer membrane, which is accompanied by the release of cytochrome c [276].

Opening of the mPTP is at least partially responsible for tissue damage after ischemia/reperfusion [278,279,280,281] in such pathological conditions as myocardial injury in ischemia [282], IR injury of the liver [283], and traumatic brain injury (TBI) [284]. AcAc and BHB prevent neuronal death in response to diamide, an activator of mPTP opening, and the effects of these ketone bodies seemed to be quite similar to the effect of CsA, a conventional mPTP inhibitor [277,285,286]. In mitochondria isolated from Kcna1^-/-^ mice with epilepsy, the KD prevents mPTP activation [287]. Interestingly, the effects of ketone bodies on the mPTP in hippocampal mitochondria were completely abolished in cyclophilin D (CypD)-deficient animals [287].

Stress including hypoxia induces fragmentation of the mitochondrial reticulum, followed by ROS generation, cytochrome C release, and culminating in cell death [288,289], and such mitochondrial fission is regulated by a dynamin-related protein 1 (Drp1) [290]. After 2 weeks of the KD, there were no significant changes in *Mfn1*, *Mfn2*, *Opa1*, or Drp1 expression in hearts before IR, unlike the fission regulator Fis1, whose expression was upregulated by ∼60% [118]. Similarly, the OGD caused significant mitochondrial fragmentation, whereas 10 mM BHB prevented the mitochondrial fission. The morphology of mitochondria of SH-SY-5Y cells treated with 10 mM BHB was similar to that of cells treated with the small-molecule mitochondrial division inhibitor (mdivi-1). Additionally, mdivi-1 inhibits Drp1, which controls mitochondrial fission. According to these findings, BHB may modulate mitochondrial dynamics by inhibiting Drp1-mediated mitochondrial fission [33].

## 7. Conclusions

Summarizing the available data, there is a huge variety of experimental models, diet compositions, and durations of intervention. However, in the majority of studies, the KD and its mimetics demonstrate beneficial cytoprotective effects. Most of the analyzed studies are related to the neuroprotective properties of KD, while the effects of the KD and BHB administration on other organs are less studied. The protective effects of the KD may be mediated through several independent pathways, the synergy of which results in sustained protection against IR injury. An increase in ketone body levels by their administration or the KD could provide HDAC inhibition and activation of the stress-responding pathways, including Sirt3-FoxO3, Nrf-2-ARE, and AMPK signaling pathways. The influence of ketone bodies on mitochondrial dynamics and control of mPTP activation may also mediate tolerance to excessive ROS generation. Meanwhile, the negative effects of high-fat and ketogenic diets were described for some organs, especially the myocardium. We consider that conflicting results require accurate separation of the effects and pathways of the KD, KD-accompanying CR, or similar high-fat diets.

## Figures and Tables

**Figure 1 ijms-24-02576-f001:**
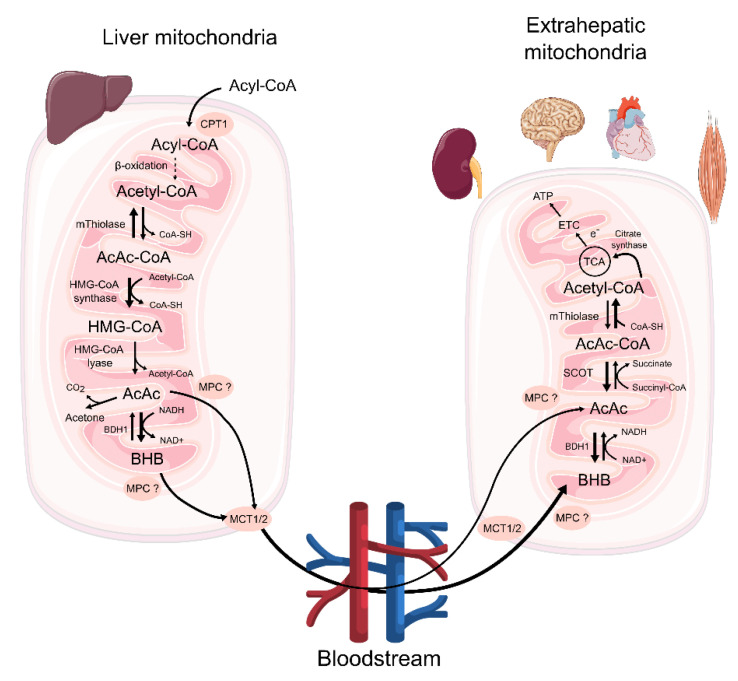
The metabolism of ketone bodies. Ketone bodies are produced in mitochondria of hepatocytes during ketogenesis, whereas utilization of ketone bodies takes place in the mitochondria of peripheral extrahepatic tissues during ketolysis. AcAc—acetoacetate; BDH1—D-β-hydroxybutyrate dehydrogenase; BHB—β-hydroxybutyrate; CPT1—carnitine palmitoyltransferase 1; ETC—electron transport chain; HMG-CoA—3-hydroxy-3-methylglutaryl-CoA; MCT1/2—monocarboxylate transporter 1/2; MPC—mitochondrial pyruvate carrier; SCOT—succinyl-CoA:3-ketoacid CoA transferase; TCA cycle—tricarboxylic acid cycle.

**Figure 2 ijms-24-02576-f002:**
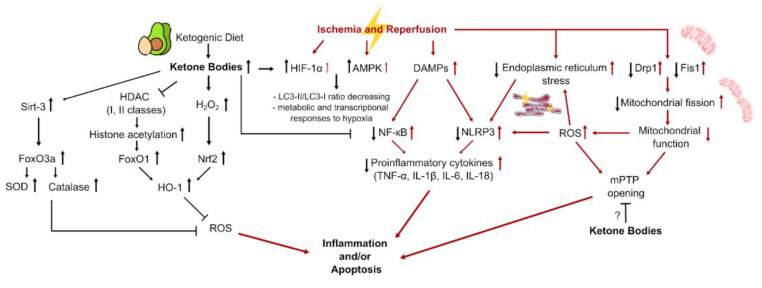
Possible mechanisms of the beneficial effects of KD and ketone bodies on IR injury. Red arrows indicate the effects mediated by ischemia and reperfusion, while black arrows specify the effects mediated by KD. AMPK—AMP-activated protein kinase; DAMPs, damage-associated molecular patterns; Drp1—dynamin-related protein 1; FoxO3a—forkhead box protein O3a; IL-(1β, 6, 18)—interleukin-(1β, 6, 18); HDAC, histone deacetylases; HIF-1α—hypoxia-inducible factor-1α; HO-1—heme oxygenase-1; mPTP—mitochondrial permeability transition pore; LC3—microtubule-associated protein 1A/1B-light chain 3; NLRP3—NOD-, LRR-, and pyrin domain-containing protein 3; NF-κB -nuclear factor-κB; Nrf2—nuclear respiratory factor 2; ROS, reactive oxygen species; SOD, superoxide dismutase; TNF-α—tumor necrosis factor-α.

**Table 1 ijms-24-02576-t001:** Studies related to the usage of KD or similar diets against IR injury.

Diet	Macronutrients, P:C:F *, %	Calorie Intake Normalization	Duration before/after Injury	Animal, Organ	Model	Positive Effect	Results	References
KD	9.5:4.7:85.8	No	14 d/28 d	Mice Hind-limb	Unilateral femoral artery ligation	− −	Long-term KD decreased blood perfusion and aggravated inflammation of ischemic tissue, delayed muscle recovery and wound healing at the surgical site, induced muscle atrophy of non-ischemic tissue.	[127]
KD	4.7:1.8:93.5	N/D **	28 d/28 d	Mice Heart	Left anterior descending artery ligation	− −	Long-term KD exacerbated cardiac dysfunction.	[128]
KD	10.4:0.1:89.5	N/D	28 d/-	Rats Brain	MCAO	+	KD-induced stabilization of HIF-1α in rat brain under normoxic conditions was associated with inflammatory response activation and neuroprotection during IR.	[129]
KD	11.3:1.5:87.2	Yes	3 d/-	Rats Kidney	Left renal artery occlusion	++	Acute KD feeding caused protein acetylation, liver AMPK activation, and increased resistance to IR-induced kidney injury. KD attenuated oxidative damage, increased antioxidant defenses, and reduced inflammation after kidney IR. KD prevented interstitial fibrosis development at two weeks, upregulation of HSP70, and chronic Klotho deficiency.	[31]
“New-KD”	12:5:84	Yes	-/70 d	Rats Brain	Controlled cortical impact	+	The new KD attenuated sensorimotor deficits, corrected spatial memory deficit, reduced the lesion size, perilesional inflammation, and oxidative damage, enhanced the mTOR pathway, and increased histone acetylation and methylation.	[32]
KD	N/D:N/D:80	No	3 d/7 d	Rats Brain	endothelin-1 injection	+	KD led to a reduction in motor behavior impairment in the model of ischemic stroke.	[130]
KD	10.4:0.1:89.5	Yes	21 d/-	Mice Brain	MCAO	+	KD provided tolerance to MCAO/R, inhibited endoplasmic reticulum stress, and suppressed TXNIP/NLRP3 inflammasome activation in the brain.	[33]
LCHFD	N/D:N/D:60	N/D	21 d/-	Mice Gut	N/D	+/−	LCHFD feeding increased gut NO levels before gut IR, but decreased them after gut IR.	[131]
KD	10.4:0.1:89.5	No	28 d/-	Mice Brain	MCAO	+++	KD was neuroprotective against focal cerebral ischemia in a concentration-dependent manner, and upregulated cytoprotective pathways associated with HIF-1α, pAKT, and AMPK.	[34]
KD	10.4:0.1:89.5	No	21 d/-	Mice Brain	MCAO	+++	KD led to a reduction in infarct volume and an increase in regional cerebral blood flow and extracellular adenosine levels in both the ischemic and the reperfusion phases. KD increased Akt and ERK1/2 phosphorylation via A1R activation, and upregulated HIF-1α/HIF-2α, VEGF, and EPO.	[35]
Fat-rich diet	17:48:35	Yes	21 d/-	Mice Brain, liver	MCAO	+/−	Fat-rich diets increased BHB levels in liver, blood, brain microdialysate, and brain homogenate 90 min after MCAO. Glucose levels were changed in the opposite manner. Reperfusion decreased BHB and increased glucose within 60 min. Citrate and succinate were moderately increased by the fat-rich diet and unchanged after stroke.	[122]
HFLCD	30:10:60	No	14 d/-	Rats Heart	IR in the isolated heart	− −	HFLCD led to an increase in free fatty acid (FFA) oxidation and a decrease in carbohydrate or ketone oxidation, both in control and IR. HFLCD led to decreased recovery of left ventricular function and reduced insulin sensitivity.	[132]
HFLCD	15:20:65	N/D	5 d/-	Mice Gut	Superior mesenteric artery occlusion	+/−	Short-term HFLCD did not affect survival after gut IR.	[133]
HFLCD	30:10:60	Yes	14 d/-	Rats Heart	Left anterior descending coronary artery ligation	− −	HFLCD did not affect nonischemic left ventricular function but led to greater myocardial injury during IR, with increased risk of death by pump failure and ventricular arrhythmias.	[118]
HFLCD	30:10:60	Yes	14 d/-	Rats Heart	IR in the isolated heart	− −	HFLCD led to increased ischemic myocardial injury, impaired recovery of function after reperfusion, and enhanced oxidative stress.	[119]
KD	10.4:0.1:89.5	N/D	21 d/-	Rats Brain	Hypoxia in hypobaric chambers	+	In the aged rats, KD improved cognitive performance under normoxic and hypoxic conditions, while motor performance remained unchanged. Capillary density and HIF-1α levels were elevated in the aged KD group independent of hypoxic challenge.	[123]
KD	10:2:78	No	25 d/-	Rats Brain	Cardiac arrest-induced cerebral ischemia	+	KD prevented cardiac-arrest-induced cerebral ischemic neurodegeneration in several brain regions.	[134]
KD	10.4:0.1:89.5	No	21 d/-	Rats Brain	MCAO	+	KD reduced infarct volumes following IR.	[135]
HFD	20:20:60	No	-/56 d	Rats Heart	Coronary artery ligation	+	HFD *** increased state 3 respiration and acyl-CoA dehydrogenase activity, but did not normalize levels of acyl-CoA dehydrogenases in IR-induced heart failure.	[125]
HFD	20:20:60	No	14 d/- or -/56 d	Rats Heart	Coronary artery ligation	+/− −	HFD following cardiac IR did not exacerbate left ventricular dysfunction and remodeling, but increased surgical mortality. HFD increased mitochondrial oxidative phosphorylation and ETC complex activity. HFD before surgery resulted in an increased surgical mortality rate	[126]
LCKD	60:10:30	No	133 d/-	Rats Heart	IR in the isolated heart	+	LCKD increased the number of mitochondria and provided a tolerance to ischemia and a faster recovery of cardiac function following reperfusion.	[124]
five experimental diets	17:21:62 ****	Yes	12 h/-	Rats Brain	MCAO	+	Infarct volumes were significantly smaller after the 1,3-butanediol diet and after the triacetin–tributyrin diet. Infarct volume correlated with the plasma glucose, but not lactate, ketone body, or acetate concentration before ischemia.	[121]

* Protein:carbohydrates:fat ratio. ** N/D—not disclosed in the paper. *** Note: this is a high-fat, but not a low-carbohydrate diet. **** Fat component was presented by one or more of the following substrates: 1,3-butanediol, triacetin, tributyrin, and long- and medium-chain triglycerides, in different ratios.

**Table 2 ijms-24-02576-t002:** Studies related to the usage of KD mimetics against IR injury.

Mimetic	Dose	Duration before/ after Injury	Animal/ Cells, Organ	Model	Positive Effect	Results	References
BHB	one intraperitoneal injection of 30 mg/kg	-/90 min	Mice	MCAO	++	Single acute BHB injection improved the neurological score determined and mitochondrial respiratory complex I and II activity after 24 h but not at later time points.	[136]
BHB sodium salt	1–100 mM	12 h/-	Mouse cardiomyocytes	Incubation under hypoxic conditions	− −	Treatment with BHB enhanced cardiomyocyte death and decreased glucose absorption and glycolysis under hypoxic conditions.	[128]
BHB	in vivo: 4 μL of BHB (250–1000 μg/kg) injected once into lateral ventricle	-/1 h	Rats Brain	MCAO	++	BHB enhanced mitochondrial respiratory chain complex I activity, reduced oxidative stress, inhibited apoptosis, improved neurological scores, and reduced infarct volume after ischemia. BHB acted through upregulation of BHB transporter SMCT1 and activation of the Erk/CREB/eNOS pathway.	[137]
	in vitro: 2–100 mM	24 h/-	Rat Neuronal cells	OGD	++		
DL-BHB	one intraperitoneal injection of 10 mmoL/kg	30 min/-	Mice Liver	Partial warm hepatic IR	+++	BHB reduced hepatocellular necrosis after IR treatment. Exogenous BHB induced acetylation of histone-3, which resulted in higher expressions of FOXO1 and HO-1 upregulation. The expression of NLRP3 in the liver and serum levels of IL-1β was suppressed by BHB.	[138]
BHB	in vivo: 8 μL/h, 1 g/mL; osmotic pumps intraperitoneally	1 d/1 d	Mice Kidney	Left renal artery and vein occlusion	+	Renal IR injury was attenuated by BHB treatment. BHB reduced the number of TUNEL-positive cells in kidney, increased expression of FOXO3, and decreased the expression of caspase-1 and proinflammatory cytokines. In an HK-2 cell line exposed to hypoxia and reoxygenation, BHB reduced cell death in a FOXO3-dependent fashion. Histone acetylation was decreased in kidneys exposed to IR and in HK-2 cells exposed to hypoxia and reoxygenation, though this effect was ameliorated by BHB through the inhibition of histone deacetylases.	[139]
	in vitro: 1–40 mM	0/12 h	HK-2 cells	OGD	+		
D-BHB sodium salt	4 doses of intraperitoneal injection of 5.0 mmol/kg	-/0–6 h	Rats Brain	Hypoxic ischemic encephalopathy	+	The BHB group demonstrated significantly lower brain pathological scores after hypoxic ischemic injury. The intact residual hemispheric and hippocampal volumes were also greater in this group. Neurological functions were unaffected.	[140]
DL-BHB	one intraperitoneal injection of 500 mg/kg	0/1 h	Rats Brain	endothelin-1 injection	++	BHB treatment reduced oxidative stress, diminished astrogliosis and neuronal death, preserved neuronal functioning, normalized perilesional perfusion, and ameliorated cerebrovascular tolerance to hypercapnia.	[141]
D-BHB	1.6 mmol/kg; osmotic pumps subcutaneously	0/1 d	Mice Heart	Left anterior descending coronary artery occlusion	+++	Treatment with BHB reduced infarct size and levels of cardiac troponin I, creatine kinase, and lactate dehydrogenase in serum, attenuated apoptosis in myocardium, and preserved cardiac function in IR mice, reduced mitochondrial formation of ROS and swelling, enhanced ATP production, partly restored mitochondrial membrane potential, and attenuated ER stress in myocardium.	[142]
BHB	10 mM	N/D *	SH-SY-5Y cells	OGD	+++	10 mM BHB prevented the mitochondrial translocation of Drp1 to inhibit mitochondrial fission, decreased ROS generation, and suppressed ER stress-induced NLRP3 inflammasome activation in OGD-injured cells.	[33]
D-BHB	0.4–1.6 mmol/kg/day; osmotic pumps subcutaneously	0/84 d	Mice Brain	Spinal cord injury	+	BHB promoted functional recovery and relieved pain hypersensitivity in mice with spinal cord injury, possibly through inhibition of histone deacetylases, suppression of NLRP3 inflammasome, and protection of mitochondrial function.	[143]
D-BHB sodium salt	7 days of twice-daily or single-dose subcutaneous injection of 1000 mg/kg	0–7 d/0	Rats Eye	Optic nerve and central retinal blood vessel transection	+++	BHB provides retinal antioxidant defense by the antioxidant Nrf2 pathway activation via modification of a fumarate metabolism.	[120]
BHB	10 mM	12 h/0	N2a cells	OGD	++	Pretreatment with BHB reduced OGD-induced cell death. Transfection with small interfering RNA (siRNA) targeting the A1 adenosine receptor before BHB pretreatment reversed the protective effect, indicating adenosine signaling pathways involved in neuronal cell tolerance to ischemic injury.	[35]
BHB+AcAc	BHB (0.4 mmol/kg), AcAc (0.45 mmol/kg), 7 doses of subcutaneous injection every hour	0/7 h	Mice Brain	MCAO	++	Ketone body treatment enhanced mitochondria function, reduced oxidative stress, and infarct volume, led to improved neurologic function after ischemia, including the neurologic scores in Rotarod and open field tests. Ketone bodies’ effects were achieved by upregulating of SIRT3 and downstream effectors, FoxO3a and SOD2, in the penumbra region.	[144]
	N/D	-	Murine neuronal cells	Rotenone treatment	++		
D-BHB sodium salt	intraventricular infusion; 10 mM 1 ul/hr	4 d/-	Rats Brain	MCAO	+	A 55–70% reduction in infarct volume was observed with BHB infusion or diet-induced ketosis. HIF-1α and Bcl-2 protein levels increased after BHB infusions. Succinate content increased 4-fold with BHB infusion	[135]
Lithium salt of AcAc	4 groups: (I) 600 mM AcAc 0.5 μL/h/14 d by osmotic pump subcutaneously; (II) daily intraperitoneal injections 250 mg/kg AcAc for 7 days; (III) intravenously 500 μL of a 200 mM AcAc; (IV) three intraperitoneal injections of 250 mg/kg AcAc	(I) 10 d/4 d; (II) 4 d/3 d; (III) 0/15 min; (IV) 0/0.5–1.5 h	Rats Brain	Iodoacetate+L-trans-pyrrolydine-2,4-dicarboxylate injection	+	AcAc efficiently protects against glutamate neurotoxicity both in vivo and in vitro.	[145]
DL-BHB	25 μmol/kg/min; intravenously using a syringe pump	60 min/-	Rats Heart	Left coronary artery occlusion	++	DL-BHB infusion after prolonged fasting reduced infarct volume and apoptosis after heart IR, possibly by increasing myocardial ATP levels.	[146]
D-BHB sodium salt	30 mg/kg/h; intravenous catheter	0/1–3 d	Rats Brain	MCAO	++	BHB reduced infarct area at 24 h, but not at 72 h after permanent MCAO. In rats with 2 h transient MCAO followed by 22 h reperfusion, BHB significantly reduced cerebral infarct area, edema formation, lipid peroxidation, and neurological deficits.	[147]
D-BHB sodium salt	3–100 mg/kg/h; intravenous catheter	30 min/0 or 0/3 h	Rats and mice Brain	N_2_-induced hypoxia in mice KCN-induced anoxia and global cerebral ischemia induced by bilateral carotid artery ligation in rats	++	BHB administered immediately after a bilateral carotid artery ligation significantly suppressed the elevation of cerebral water and sodium contents as well as maintaining high ATP and low lactate levels. BHB demonstrated protective effects on cerebral hypoxia, anoxia, and ischemia-induced metabolic change.	[148]
1,3-Butanediol	from 1 to 4 intraperitoneal injections injections; 25 mmol/kg	30 min/0–9 h	Rats Brain	Embolization by microspheres	++	1,3-Butanediol attenuated ischemia-induced metabolic changes by increasing the concentrations of phosphocreatine, ATP, and glycogen and by reducing the concentrations of pyruvate and lactate.	[149]
D-BHB L-arginine salt	20 μmol/kg/min; intravenous catheter	-/90 min after 40 min occlusion	Dogs Heart	Left anterior descending coronary artery occlusion	+	The BHB treatment stabilized the left ventricular function.	[150]
BHB	one 12 mg/mouse intravenous or 30–120 mg/mouse intraperitoneal injection	30 min/0	Mice Global hypoxia	N_2_-induced hypoxia	++	BHB with glucagon combination increased survival time in global hypoxia.	[151]

* N/D—not disclosed in the paper.

## Data Availability

The data that support the findings of this study are available from the corresponding author upon reasonable request.

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
