# Peer review of "Ketogenic Diet and Ketone Bodies against Ischemic Injury: Targets, Mechanisms, and Therapeutic Potential"

_ijms, 2023, doi:10.3390/ijms24032576_

Round 1
Reviewer 1 Report
Makievskaya et all summaries the effect of ketogenic diet and ketone bodies on ischemic injury. Authors starts the review by providing a brief overview of ketone body metabolism, followed by a summary of various ketogenic diet and ketone body mimetic. This review summarizes the studies that focused on the effect of ketone bodies or ketogenic diet on various ischemia reperfusion injury and provides a comprehensive overview of potential mechanisms. This review will be well received since it displays both pros and cons of ketogenic diet, its limitations and caveats.
Few smaller points to consider:
- Figure 1 suggests that MCT1/2 is transporting ketones through mitochondria, where MCT are rather localized on the cell surface. Can authors provide a reference that support this notion?
- Page 2, line 63: authors suggest many amino acids in addition to Leu and Lys are ketogenic (Phe, Ile, Thr, Trp, Tyr), can authors provide a reference to support this claim?
- Could authors provide explanations of the abbreviations used in each Figures?
Author Response
We thank the reviewer for valuable comments. We have made the appropriate changes in the text and the figures to address all questions. We appreciate the constructive criticism and hope that after these revisions, our manuscript will not raise more critique. Below, we present the reviewer’s specific comments (in black) with our replies (in blue).
Reviewer 1
Makievskaya et all summaries the effect of ketogenic diet and ketone bodies on ischemic injury. Authors starts the review by providing a brief overview of ketone body metabolism, followed by a summary of various ketogenic diet and ketone body mimetic. This review summarizes the studies that focused on the effect of ketone bodies or ketogenic diet on various ischemia reperfusion injury and provides a comprehensive overview of potential mechanisms. This review will be well received since it displays both pros and cons of ketogenic diet, its limitations and caveats.
Few smaller points to consider:
- Figure 1 suggests that MCT1/2 is transporting ketones through mitochondria, where MCT are rather localized on the cell surface. Can authors provide a reference that support this notion?
We thank the reviewer for the valuable comment. The reviewer is absolutely right that MCT1 and MCT2 are the carriers located on the cell surface (DOI: 10.2220/biomedres.36.279). We have corrected Figure 1 and moved MCT1/2 transporters symbol to the cell membrane. Of note, the mechanisms through which ketone bodies are transported across the mitochondrial inner membrane are still unclear (doi:10.1016/j.cmet.2016.12.022). One of the putative mechanisms of such transport is mitochondrial pyruvate carrier (MPC) suggested as one of the ways of β-hydroxybutyrate entering mitochondria (doi:10.1007/s11064-016-2099-2). We have added this information to the text of the manuscript and Figure 1.
- Page 2, line 63: authors suggest many amino acids in addition to Leu and Lys are ketogenic (Phe, Ile, Thr, Trp, Tyr), can authors provide a reference to support this claim?
Thank you for the recommendation, we have supported this statement with the appropriate reference (doi: 10.1016/b978-0-12-383864-3.00013-2, see page 388).
- Could authors provide explanations of the abbreviations used in each Figures?
We have provided explanations of the abbreviations in each Figure legend.
Reviewer 2 Report
Makievskaya et al studied the mechanisms of effects of KDs and ke
tone bodies on ischemic damage to various organs in it. It can be valuable study in its field. The English used is correct and readable. There are appropriate and adequate references to related and previous work. However, I have some main comments:
Major:
The introduction is so short. It must be improved in many cases. The gap of the subject, the solutions, the previous background....
The search methodology should be added.
The sections should be re-organized to more detailed and clear sections.
Minor:
please also use some 2022 references.
Author Response
We thank the reviewer for valuable comments. We have made the appropriate changes in the text to address all questions. We appreciate the constructive criticism and hope that after these revisions, our manuscript will not raise more critique. Below, we present the reviewer’s specific comments (in black) with our replies (in blue).
Reviewer 2
Makievskaya et al studied the mechanisms of effects of KDs and ke
tone bodies on ischemic damage to various organs in it. It can be valuable study in its field. The English used is correct and readable. There are appropriate and adequate references to related and previous work. However, I have some main comments:
Major:
The introduction is so short. It must be improved in many cases. The gap of the subject, the solutions, the previous background....
We thank the reviewer for this recommendation; we have expanded the Introduction section to clarify the background and the aims of our study.
The search methodology should be added.
Thank you for the useful suggestion.
Basic search was performed on PubMed and Google Scholar using the following search terms: “((ketogenic diet) OR (KD)) AND ((ischemia) OR (ischemia-reperfusion))”, “((beta-hydroxybutyrate) OR (BHB)) AND ((ischemia) OR (ischemia-reperfusion))”, “((acetoacetate) OR (AcAc)) AND ((ischemia) OR (ischemia-reperfusion))” and “(ketone bodies) AND ((ischemia) OR (ischemia-reperfusion))”. Additional keywords related to IR injury were added for specific pathologies search, i.e., "ischemic stroke" or "myocardial infarction". All the original articles from 1980 to 2022, with available abstract and full texts, were included.
Inclusion criteria were: (1) original research articles; (2) study focused on the usage of KD, similar diets or its mimetics against IR injury.
Exclusion criteria were: (1) reviews, meta-analyses, case studies, case reports, comments, study protocols, letters; (2) studies focusing on effects of interventions that induce endogenous ketosis such as CR; (3) articles not written in the English language.
References in the selected articles were analyzed to identify any other relevant studies that were not included during the previous searches.
We have added a brief description of search methodology to the introduction section.
The sections should be re-organized to more detailed and clear sections.
Thank you, we have revised some section titles and re-organized chapters to make the structure of the review more clear.
Minor:
please also use some 2022 references.
To address this comment, we have additionally analyzed the most recent studies about ketogenic diet and added the found issues to the Table 1 and Table 2, thank you.
New references [128], [129], [130], [155], [173] have been added to the references list, in addition to 2022 references [31], [93], [169], [184], [185], [213], [233], [267].
Round 2
Reviewer 2 Report
The modifications are good. It can be accepted.